# Systematic review of effect of data augmentation using paraphrasing on named entity recognition

**Saket Sharma**[1][¶]     **Aviral Joshi**[1][¶]     **Namrata Mukhija**[1][§][*]     **Yiyun Zhao**[¶]

**Hanoz Bhathena**[¶]     **Prateek Singh**[¶]     **Sashank Santhanam**[‖][†]     **Pritam Biswas**[¶]

[¶]Machine Learning Center of Excellence, JPMorgan Chase
[§]New York University
[‖]Apple
`saket.sharma@jpmchase.com`

## Abstract

While paraphrasing is a promising approach for data augmentation in classification tasks, its effect on named entity recognition (NER) is not investigated systematically due to the difficulty of span-level label preservation. In this paper, we utilize simple strategies to annotate entity spans in generations and compare established and novel methods of paraphrasing in NLP such as back translation, specialized encoder decoder models such as Pegasus, and GPT-3 variants for their effectiveness in improving downstream performance for NER across different levels of gold annotations and paraphrasing strength on 5 datasets. We also analyze the quality of generated paraphrases based on their entity preservation and paraphrasing language quality. We find that the choice of the paraphraser greatly impacts NER performance, with one of the larger GPT-3 variants exceedingly capable at generating high quality paraphrases, improving performance in most cases, and not hurting others, while other paraphrasers show more mixed results. We also find inline auto annotations generated by larger GPT-3 to be strictly better than heuristic based annotations. We find diminishing benefits of paraphrasing as gold annotations increase for most datasets. While larger GPT-3 variants perform well by both entity preservation and human evaluation of language quality, those two metrics do not necessarily correlate with downstream performance for other paraphrasers.

## 1 Introduction

Named entity recognition (NER) seeks to extract entity mentions (e.g., Shakespeare, Warwickshire) from a text (Shakespeare was born and raised in Warwickshire) for predefined categories of interest (such as people and locations). It is a critical component underpinning many industrial pipelines for a variety of downstream natural language processing applications such as search, recommendation, and virtual assistant systems. However, in real-world applications, there is often a scarcity of labeled data for training advanced deep neural models because span-level NER annotations are costly, and domain expertise may be needed to annotate data from domains such as finance, biomedical sciences, etc.

Data augmentation is often used as an alternative to address the data scarcity issue in many NLP tasks (see an NLP data augmentation survey by Feng et al. (2021)). However, data augmentation

---

[*]Work done while interned at JPMorgan Chase.
[†]Work done while at JPMorgan Chase.
[1]Equal contribution.

NeurIPS 2022 Workshop on Synthetic Data for Empowering ML Research.

for NER imposes additional challenges because NER requires token/span level label preservation. Therefore, most existing works on NER data augmentation primarily focus on local replacement for entity mentions (Dai and Adel, 2020; Zhou et al., 2022; Liu et al., 2022; Zhu et al., 2021) as well as context words (Dai and Adel, 2020; Li et al., 2020). The replacements can be other mentions with the same labels (Dai and Adel, 2020), synonyms from an external lexical resource such as wordnet (Dai and Adel, 2020), or tokens generated by the pretrained language models such as BERT via masked token task (Zhou et al., 2022; Liu et al., 2022; Zhu et al., 2021). However, to enhance the reliability of masked token prediction, the language model usually needs to be fine-tuned on the NER training data and label information is often inserted close to the [MASK]s (Zhou et al., 2022; Zhu et al., 2021), which requires a decent amount of labeled training data.

This work primarily focuses on the less-studied data augmentation method for NER – paraphrasing – which has the potential to introduce structural and lexical replacement and does not assume many labeled examples. Specifically, we compare established, and novel paraphrasing methods and propose simple ways to preserve span-level labels. Unlike most existing studies that focus on the influence of the amount of gold data only, we systematically investigate the effect of different levels of paraphrasing on downstream performance, at different levels of gold annotations across 5 datasets. We investigate the quality of paraphrases from 6 different systems as augmentation data, as well as stand alone training data for NER; and also conduct quality analysis of paraphrases generated by different systems based on the NER preservation and language quality.

We find paraphrasing to be generally effective in low data regimes for most paraphrasers. However, the choice of paraphrases affects the magnitude, and direction of the change in performance across all levels of gold data. We find the use of LLMs to generate inline annotations[3] while paraphrasing to be superior to simpler heuristics, and GPT-3 Davinci variant with inline annotations to be a generally superior choice across datasets. While paraphrases generated by GPT-3 with inline annotations also score highest on NER preservation and language quality, we do not see broader correlation between paraphrase quality metrics and downstream performance.

## 2 Datasets and Paraphrasers

### 2.1 Datasets

NER datasets are chosen to have coverage across a variety of domains including news, Wikipedia, Twitter, biomedical research and search; while also having a diverse set of entity types (word phrases, alphanumeric, datetime, alphabetical etc.).

We choose 5 datasets based on the above principles: Ontonotes5 (Hovy et al., 2006), Tweebank (Jiang et al., 2022), WNUT 2017 (Derczynski et al., 2017), MIT Restaurant NER dataset (MIT-R) (Liu et al., 2013), BioCreative V CDR (BC5CDR) (Wei et al., 2016). Pre-formatted versions of all datasets are sourced from the TNER project (Ushio and Camacho-Collados, 2021) on Huggingface datasets(Wolf et al., 2020). Datasets such as WNUT also have rare entities by design, allowing us to probe robustness against entity memorization.

### 2.2 Paraphrasers and postprocessing

In our experiments, we compare six paraphrasing systems:(1) Back Translation, (2) Pegasus, (3) Ada (Prompt A), (4) Ada (Prompt B), (5) Davinci (Prompt A) and (6) Davinci (Prompt B). We generate a maximum of 4 unique paraphrases per seed gold sentence for each paraphraser and postprocess the paraphrases with simple re-annotation and filtering.

#### 2.2.1 Back-translation; BT

Back translation has been widely used as a data augmentation method (Sugiyama and Yoshinaga, 2019; Corbeil and Ghadivel, 2020; Xie et al., 2020) including in phrase based systems like Bojar and Tamchyna (2011). For our experiments we use pre-trained English-German and German-English models available from Huggingface Hub [4] via Tiedemann and Thottingal (2020) and the model

---

[3]Inline annotation: [Shakespeare](PERSON) was born and raised in [Warwickshire](LOC)

[4]https://huggingface.co/models

| | MIT-R | Onto-notes | BC5-CDR | Twee-bank | Wnut-17 |
|---|---|---|---|---|---|
| BT | 1 | 0 | 2 | 0 | 3 |
| Pegasus | 1 | 0 | 13 | 3 | 8 |
| Ada-A | 10 | 0 | 0 | 11 | 0 |
| Ada-B | 4 | 0 | 0 | **16** | 2 |
| DaV-A | 3 | 0 | 4 | 5 | 0 |
| DaV-B | **26** | **35** | **26** | 10 | **27** |

Table 1: Counts the configurations of G & P where a paraphraser shows highest improvement over no paraphrasing baseline for a given G. DaV-B short for DaVinci (B) outperforms other paraphrasers across most datasets

| | MIT-R | Onto-notes | BC5-CDR | twee-bank | Wnut-17 |
|---|---|---|---|---|---|
| BT | 0.66 | 0.74 | 0.76 | **0.41** | 0.30 |
| Pegasus | 0.68 | 0.75 | 0.78 | 0.33 | 0.23 |
| Ada-A | 0.71 | 0.73 | 0.74 | 0.36 | 0.23 |
| Ada-B | 0.70 | 0.72 | 0.74 | 0.34 | 0.23 |
| DaV-A | 0.67 | 0.75 | 0.76 | 0.39 | 0.27 |
| DaV-B | **0.73** | **0.80** | **0.82** | **0.41** | **0.32** |

Table 2: Test micro-F1 when training using only paraphrases with P=1 for full dataset. Number in bold is the maximum for a given dataset. DaV-B short for DaVinci(B) outperforms all paraphrasers across datasets

architecture used is BART (Lewis et al., 2019). We use a temperature parameter of 0.8 with greedy decoding.

### 2.2.2 PEGASUS Paraphraser

PEGASUS, introduced in Zhang et al. (2020) for the purpose of summarization, is a large (568mn parameters) pre-trained transformer (Vaswani et al., 2017) based encoder-decoder model, pre-trained using a custom self-supervised objective. To use it as a paraphraser the model was fine-tuned on a paraphrasing task. We use an off-the-shelf version of PEGASUS fine-tuned for paraphrasing released on Huggingface model hub [5]

### 2.2.3 GPT-3 variants

GPT-3 Brown et al. (2020) is an auto-regressive decoder only transformer pre-trained for language modeling, showing impressive in-context learning, and instruction following ability ((Radford et al., 2019); (Sanh et al., 2021); (Wei et al., 2021); (Ouyang et al., 2022), (Campos and Shern, 2022)). We use the OpenAI API [6] to query the Ada (∼350M parameters), and DaVinci (∼175B parameters) variants of GPT-3. We prompt both GPT-3 variants with two versions of one shot prompts with a temperature of 0.8, max length of 100, and default values for other parameters (See Appendix 6.1):

- Prompt A – GPT-3 variant is instructed to generate paraphrases without specific instruction to retain inline annotation for entities

- Prompt B – GPT-3 variant is instructed to generate paraphrases, while also retaining inline annotation for entities

**Post-processing & filtering of paraphrases** We re-annotate outputs of all paraphrasers based on a case insensitive exact match search for the entity values present in seed sentence. In the case of LLMs generating inline annotations, this logic is used to supplement annotations generated by the model, relying on the model generated annotations in cases of conflicts. Further filtering is applied to the paraphrases from all models to remove paraphrases for seed sentences shorter than 15 characters, remove paraphrases that are a duplicate of the seed sentence or of another paraphrase, and when generation contains an entity not present in entity space of the dataset. We also retain only the first generation of multiline generations for paraphrasers generating a numbered list of paraphrases (common with prompt driven GPT-3 variants Appendix 6.2)

---

[5]https://huggingface.co/tuner007/pegasus_paraphrase
[6]https://beta.openai.com/

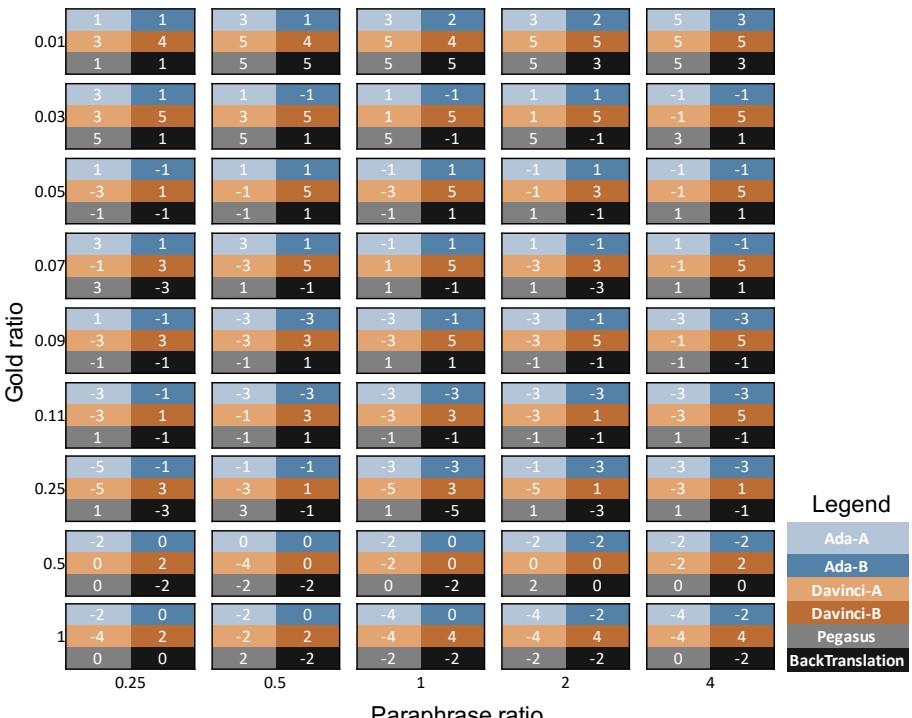

Figure 1: Matrix of scores of how F1 changed relative to the no paraphrasing (P=0) baseline after the addition of synthetic data across datasets for different G & P ratios. Improvement/worsening in any dataset at a given G/P ratio gets a score of +1/-1 respectively, and aggregation is then done across datasets. Higher numbers represent better performance across datasets. Score $\in [-5, 5]$

## 3 Experiments

### 3.1 Using paraphrases as augmentation data

**Experimental setup** In real world scenarios, we get annotated gold data incrementally. In our experiments, we simulate this by sampling gold data at various levels (1% to 100%) by building upon previous samples. For example, while generating gold sample for G=0.01 (first sample), we sample 1% of the total dataset, stratified by entities. However, when sampling for G=0.02, we retain the sample from the first step, and sample an additional 1% of the remaining dataset. This process is repeated until G=1. As a result, going from G=0.25 to G=0.5 does not actually double the gold dataset used in training. At each sampling step, we also sample an equivalent percentage of gold samples with no entities. Early experiments suggested increased benefit of paraphrasing at lower dataset size, so we explore more G ratios in this space. Additionally we only go upto G=0.25 for the really large Ontonotes dataset for speed.

For each gold to paraphrase ratio combination, we first sample gold data by the method described above. Then we randomly sample paraphrases for the gold IDs sampled until and including the current sample. This dataset is used to fine-tune a distilbert-base-cased base model for named entity recognition using the 1-step training described by (Okimura et al., 2022) using standard classification loss over hidden states of individual tokens. The models are trained to convergence with early stopping with a patience of 5.

We generate the overall, and entity specific micro F1 for each G/P combination along with standard deviation across three runs.

**Results** As Table 1 and Figure 1 suggest, choice of paraphraser strongly dictates the augmentation performance. GPT-3 Davinci (Prompt B) consistently outperforms, or matches other paraphrasers and is a safe default choice for paraphrasing across domains. Across the Davinci variants, inline annotations with Prompt B strictly outperform those introduced using heuristics. Davinci Prompt B

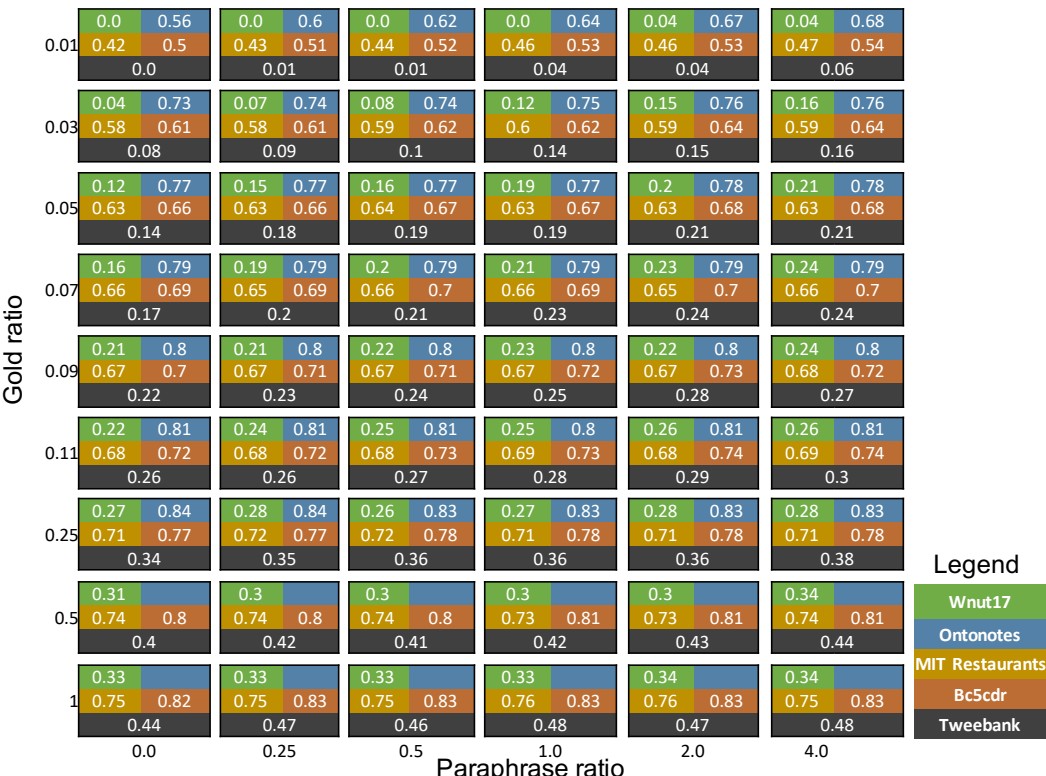

Figure 2: Micro F1 for Davinci (Prompt B) on datasets across gold and paraphrase ratios

also achieves or matches best performance at G=1 (0.25 for Ontonotes) and P=4 across all datasets. Ada variants show the most inconsistent results, with Backtranslation and Pegasus outperforming them as well as Davinci (Prompt A) in many cases. Full results are available in Appendix 6.4.

**Increasing gold ratios**  While we run similar experiments on all paraphraser-dataset pairs, we share the results from DaVinci (Prompt B) on all datasets in Figure 2 because of its consistent improvements in NER performance. (Full results Appendix 6.4). We see consistent benefits of paraphrasing at lower gold ratios, and diminishing returns in relative performance bump as we go to higher values. Other paraphrasers show similar trends at low G ratios with some exceptions (Ada variants in BC5CDR, and Backtranslation on MIT-R) (See Figure 1, Appendix 6.4), although we see a lot more mixed results at medium to high G ratios.

**Increasing paraphrase ratios**  For DaVinci (Prompt B), as we increase paraphrasing, an initial bump in performance is seen followed by flattening performance, with a minor drop in performance in some cases (Figure 2). Other paraphrasers show more mixed results (Appendix: 6.4)

Finally, paraphrasing with DaVinci (Prompt B) also leads to improvements in performance for the WNUT17 dataset (Figure 2, Appendix 6.4) across all G ratios, showing signs of robustness against entity memorization, although deeper analysis on memorization is left for future work.

### 3.2  Using paraphrases as training data

**Experimental setup**  We further evaluate quality of paraphrases directly by using **only** synthetic data to train NER models. These experiments are done only for P=1 for paraphrases generated for all training data (G=1).

**Results**  As seen in Table 2, we find GPT-3 DaVinci Prompt B paraphrases performing best across all datasets. The trends among paraphrasers track augmentation performance in Figure 1 and Appendix 6.4.

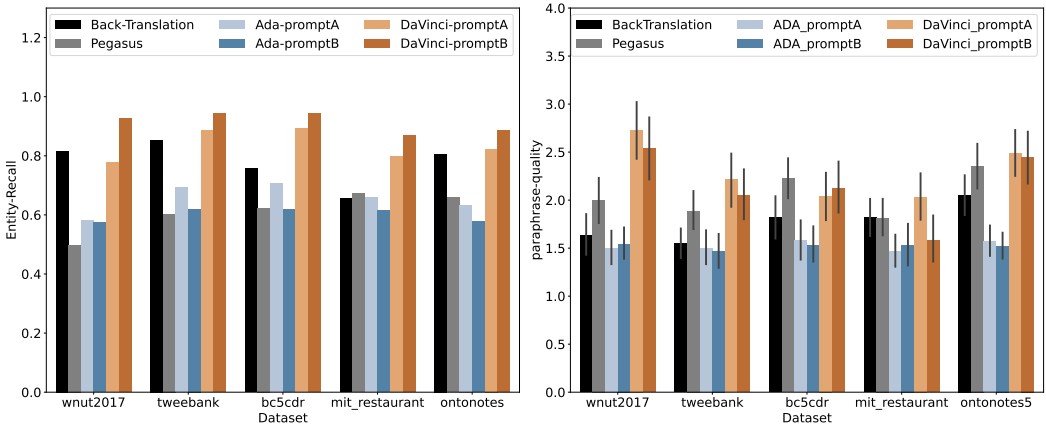

(a) Entity recall evaluation        (b) Human evaluation of paraphrasing quality

Figure 3: Paraphrase Evaluation

### 3.3 Paraphrase generation quality Analysis

Besides assessing usefulness for NER with actual training, we investigate paraphrase generation quality directly from two perspectives – entity preservation and paraphrase quality to see to what extent these metrics correlate with NER performance.

As entities are central to NER, we hypothesize entity preservation to be important for performance. We count the number of gold entities that appear in paraphrases with correct annotations via a case insensitive string match (entity recall). This calculation sets a lower bound of the entity preservation accuracy.

Good paraphrases are also expected to introduce form variety while preserving the meaning faithfully, potentially helping downstream performance. We asked three human annotators to annotate paraphrases generated by the six systems for 50 training examples sampled for each dataset. Specifically, human annotators were instructed to ignore the entity accuracy and to score paraphrases from 1-5 based on the paraphrasing quality. Our guidelines are similar to (Niu et al., 2020) (Appendix 6.5)

#### 3.3.1 Results

According to Figure 3(a), among all the paraphrase systems Davinci (Prompt B) has the highest entity recall rate, followed by Davinci (Prompt A) and backtranslation. While, Ada and Pegasus are more likely to lose gold entities. This suggests a large-sized GPT-3 model with an appropriate prompt can generate examples with high-quality inline entity annotations but a small-sized GPT-3 consistently underperforms even a simple Back-translation system. Figure 3(b) shows Davinci systems always have the best human evaluation scores across datasets followed by Pegasus and Back-translation, while Ada systems are consistently the worst.

In summary, we find that paraphrases generated by the Davinci (Prompt B) system often preserve entities and are of a good paraphrasing quality whereas Ada systems consistently underperform other systems in both metrics across datasets. These results are partially consistent with the downstream evaluations in that the augmentation data generated by Davinci (Prompt B) have reliably better downstream performance compared to other systems. However, broader trends in paraphrasing quality do not track with downstream NER performance.

## 4 Future work

While our work proposes a paraphrasing pipeline that performs consistently better than established paraphrasing pipelines in NER, we expect further benefits to come from more exhaustive tuning

of prompts used to generate paraphrases. Another potential direction to improve downstream performance is to explore better (than random) sampling strategy for paraphrases (based on entity density, entity recall, or other metrics).

## 5 Conclusion

We study the effect of six paraphrasing systems on downstream NER performance across 5 datasets. We find that the choice of paraphraser system (model + prompt) strongly affects NER performance. GPT-3 Davinci (Prompt B) performs the best at both NER performance and paraphrasing quality metrics but other paraphrasers show mixed results, suggesting that GPT-3 Davinci (Prompt B) is a strong default choice. We further find that generating inline annotations using GPT-3 Davinci works superior to strictly heuristic based annotations. While we find paraphrasing to be more effective at lower amount of training data, it helps at higher levels depending on dataset, and paraphraser. We observe dataset dependent optimal paraphrase ratios for Davinci (Prompt B), with diminishing results as paraphrasing is increased; whereas other paraphrasers show mixed results. Paraphrases from Davinci (Prompt B) have the best quality, and downstream performance, but we do not find general correlation between paraphrase quality, and downstream performance.

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

# 6 Appendix

## 6.1 Prompt design

The following prompts are used in the experiments:

```
Create a paraphrase for inputs like the following example:

Input: Japanese band The Altruists is releasing their hit single this fall.
Paraphrases:
1. The Altruists, a Japanese band is releasing their hit single this fall.

Input: {PROMPT_FILLER}
Paraphrases:
1.
```

Figure 4: GPT-3 is instructed to generate paraphrases similar to an example without any specific instruction to retain inline annotations

```
Create a paraphrase for inputs like the following example. Preserve the annotations in the [] and ():

Input: Japanese band [The Altruists](ORG) is releasing their hit single this fall.
Paraphrases:
1. [The Altruists](ORG), a Japanese band is releasing their hit single this fall.

Input: {PROMPT_FILLER}
Paraphrases:
1.
```

Figure 5: GPT-3 is instructed to generate paraphrases similar to an example, asking it to retain inline annotations

## 6.2 Multiline generation

LLM paraphrasers can be triggered to generate multi-line outputs. This behavior is more common in Ada variants over DaVinci, showing the DaVinci is better at following prompt instructions.

```
Create a paraphrase for inputs like the following example:

Input: Japanese band The Altruists is releasing their hit single this fall.
Paraphrases:
1. The Altruists, a Japanese band is releasing their hit single this fall.

Input: #Volunteers are key members of #CHEO's One Team – helping kids and families be their healthiest #NVW2016 URL1387
Paraphrases:
1. The #Volunteers are key members of #CHEO's One Team - helping kids and families be their healthiest for #NVW2016.

2. The #Volunteers are key members of #CHEO's One Team - helping kids and families be their healthiest for #NVW2016.

3. The #Volunteers are key members of #CHEO's One Team - helping kids and families be their healthiest for #NVW2016.
```

Figure 6: GPT-3 variants sometimes generate multiple numbered paraphrases. We choose to retain only the first paraphrase in these cases

## 6.3 Detailed results across different gold data sizes for all datasets

### 6.3.1 BC5CDR

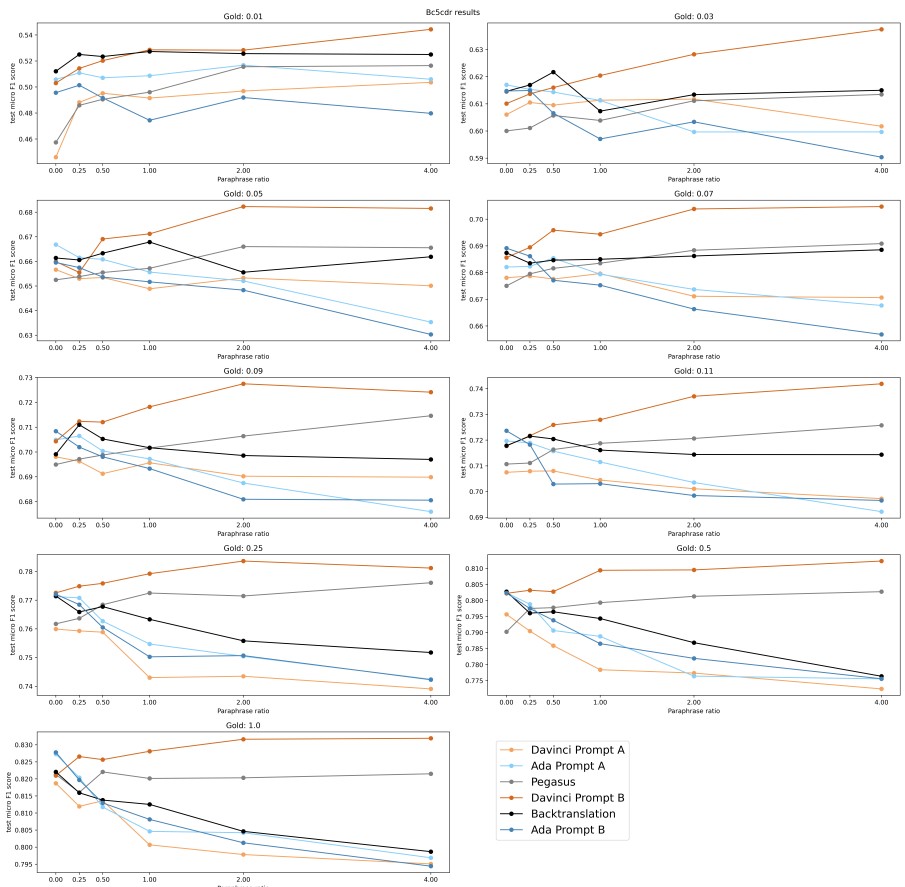

### 6.3.2 Ontonotes

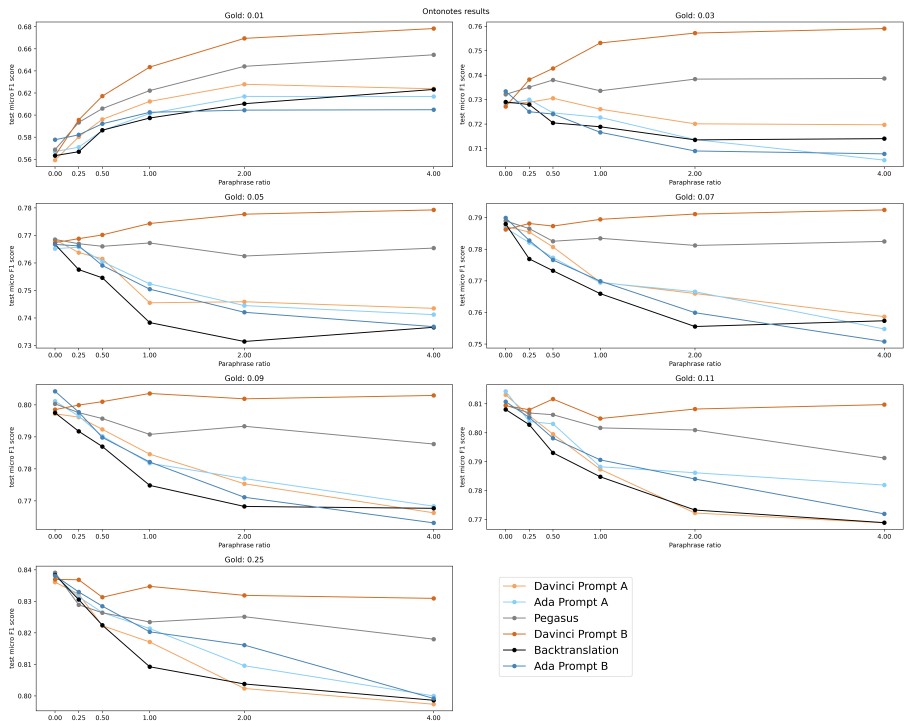

### 6.3.3 MIT-R

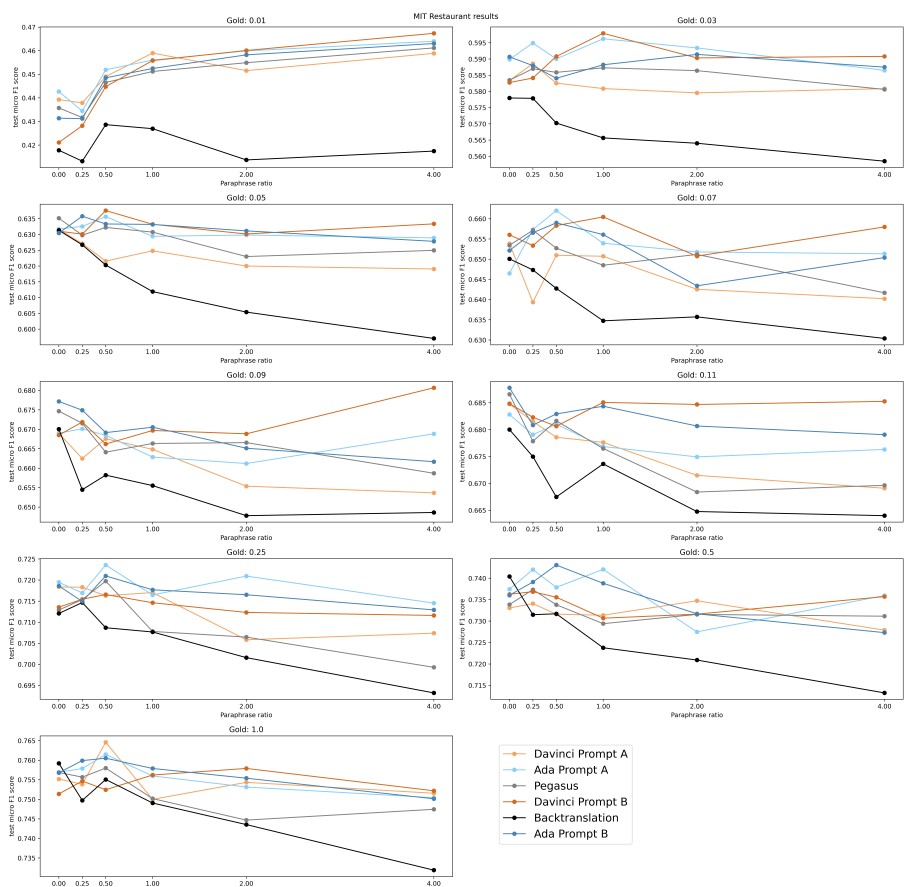

### 6.3.4 Tweebank

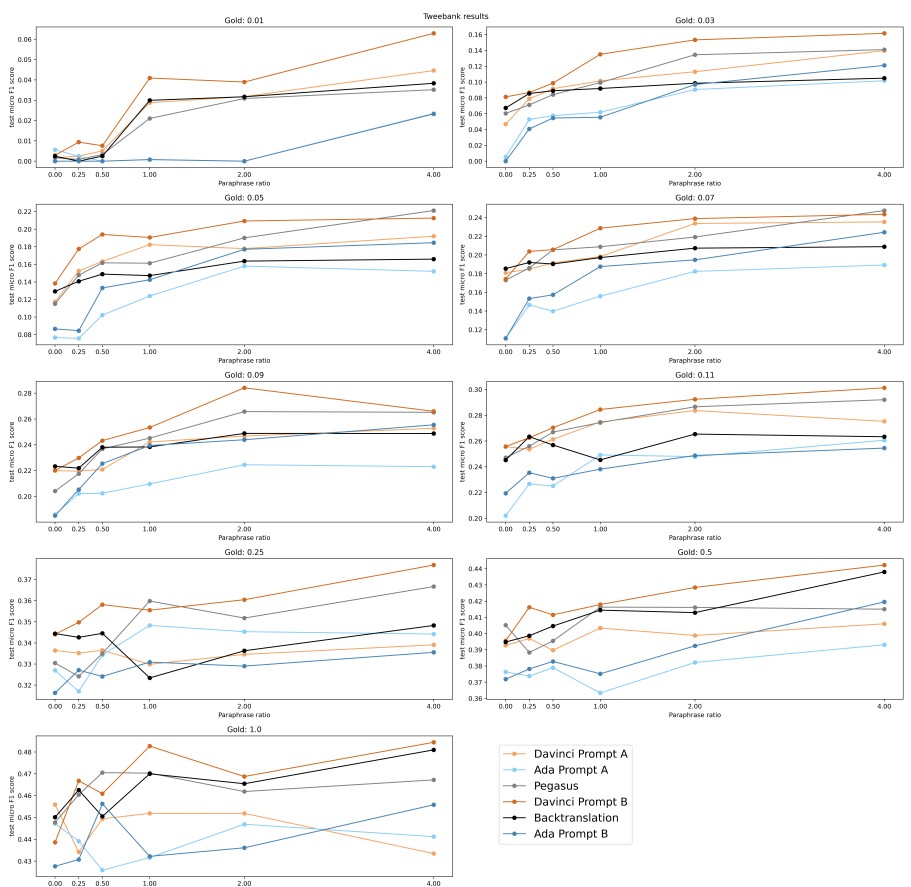

## 6.3.5   WNUT-17

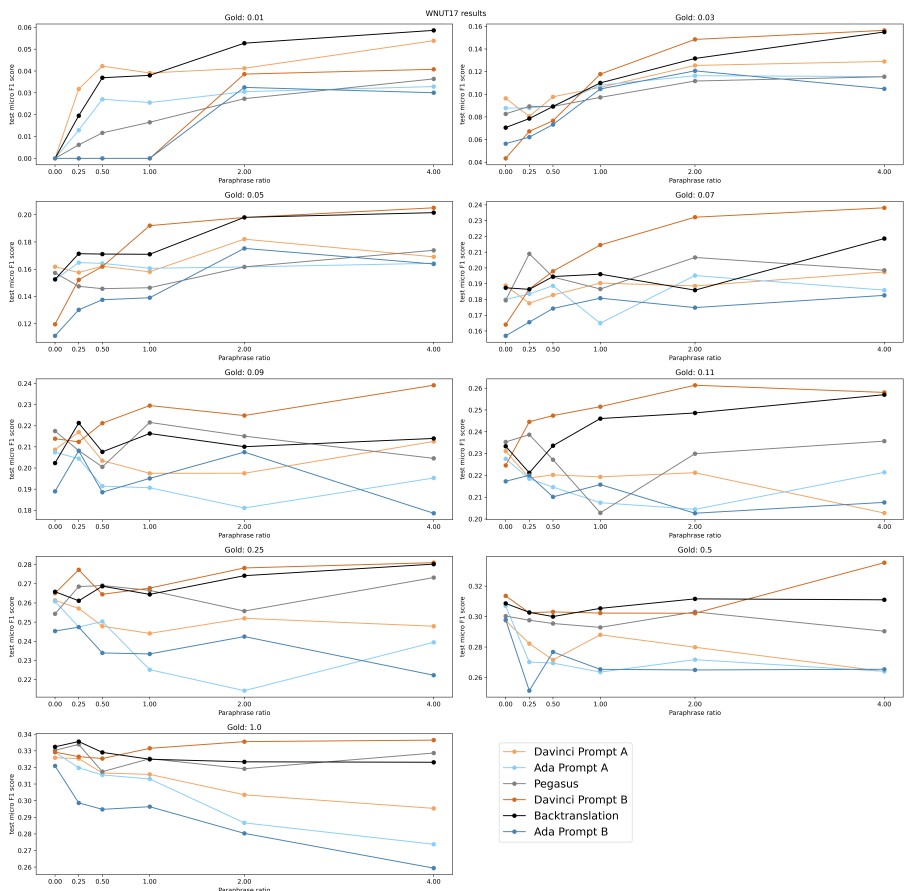

## 6.4 Heatmap of micro-f1 scores across all datasets & paraphrasers

### 6.4.1 BC5CDR

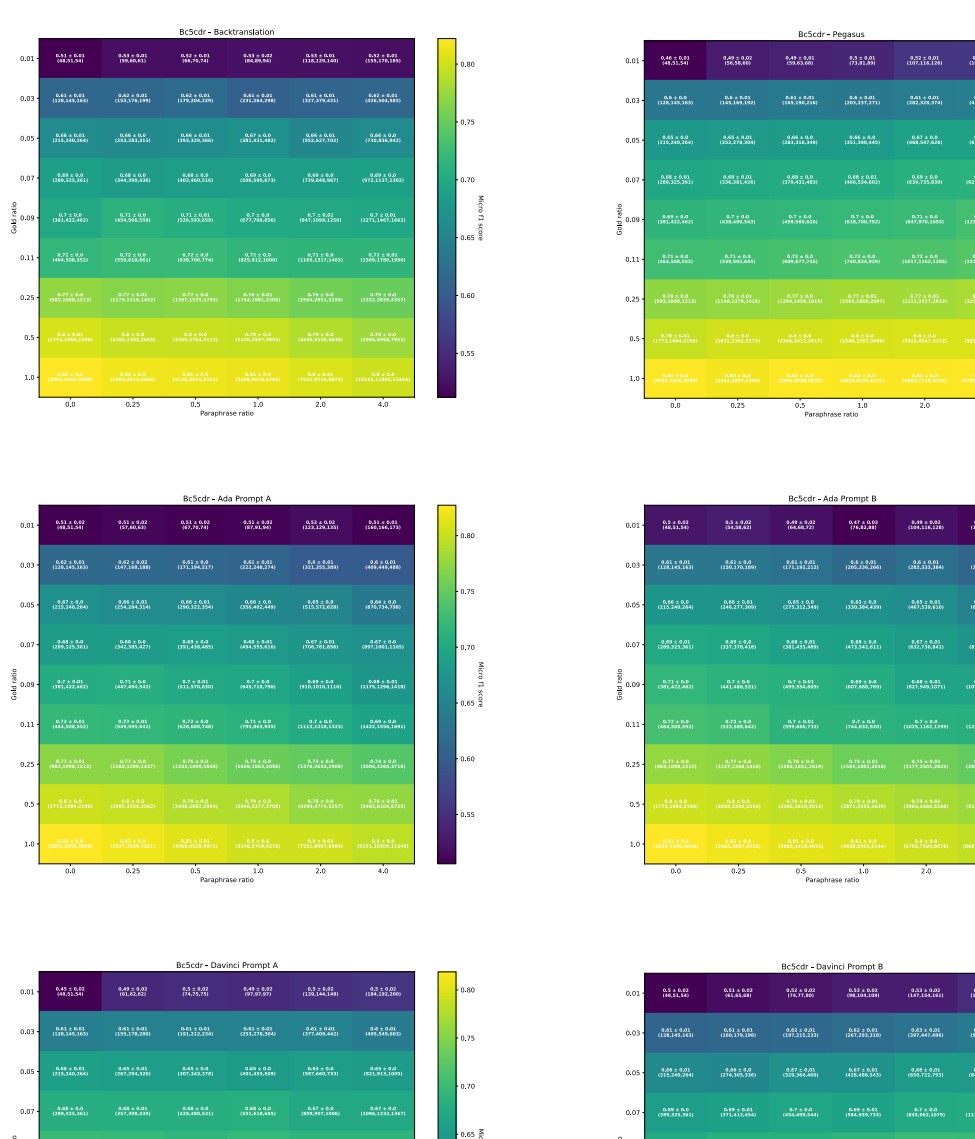

### 6.4.2 Ontonotes

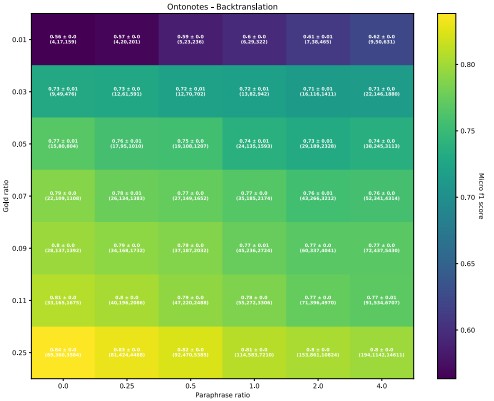

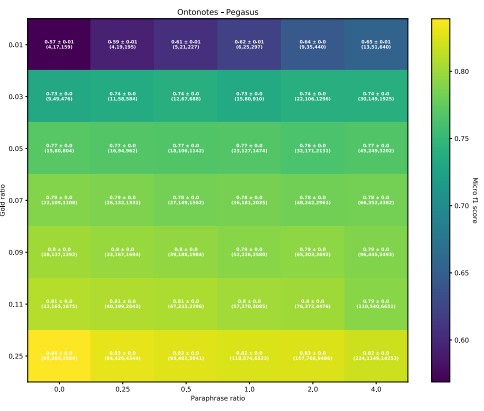

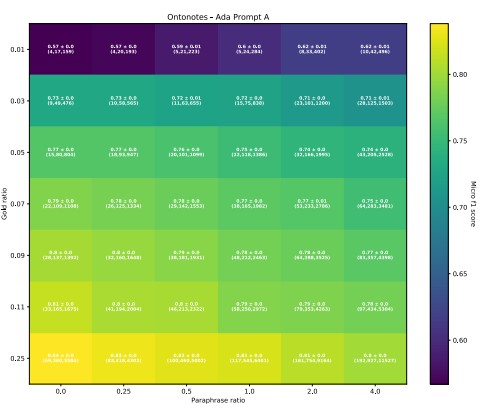

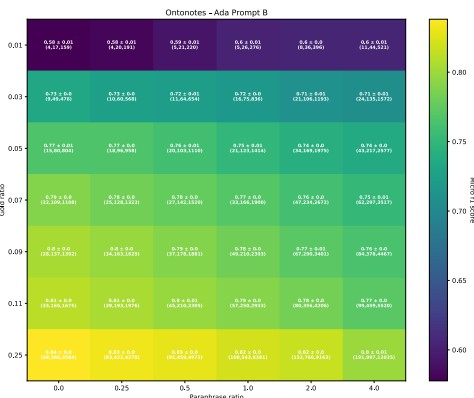

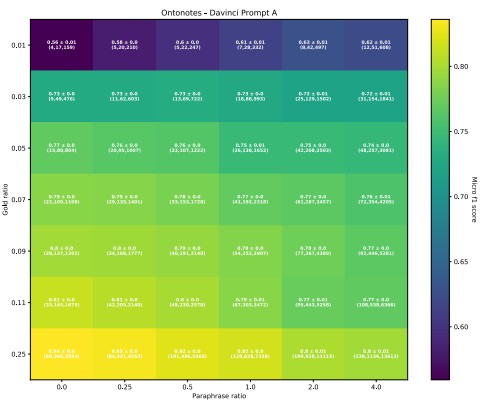

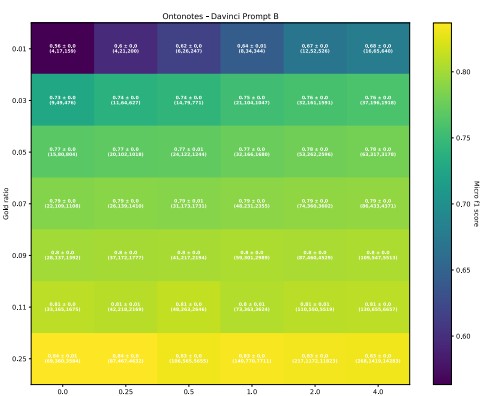

### 6.4.3 MIT-R

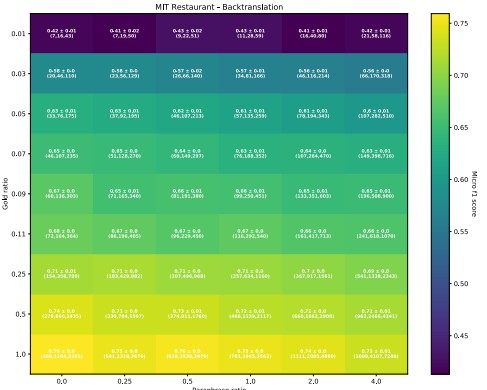

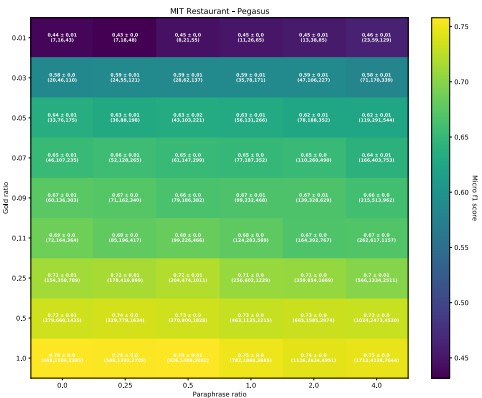

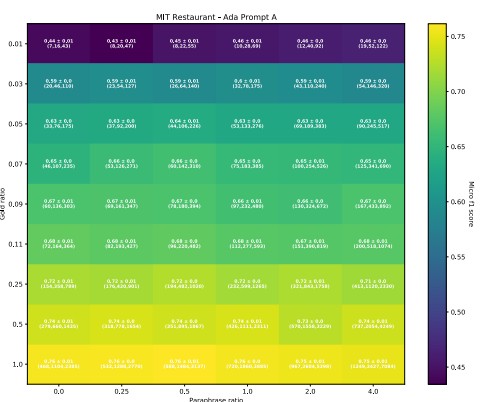

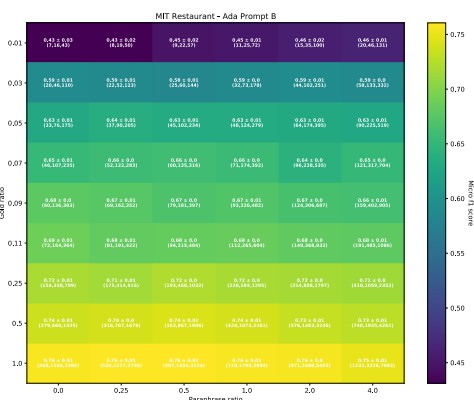

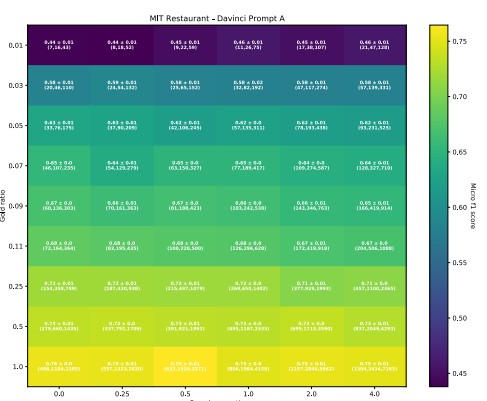

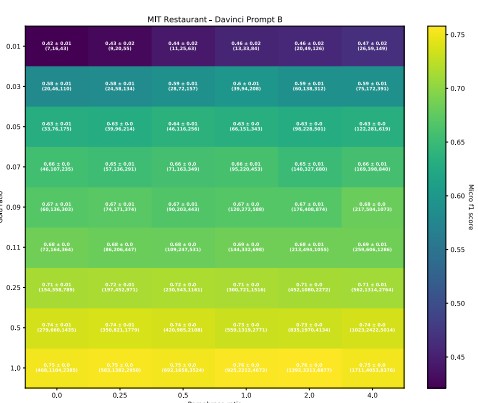

### 6.4.4 Tweebank

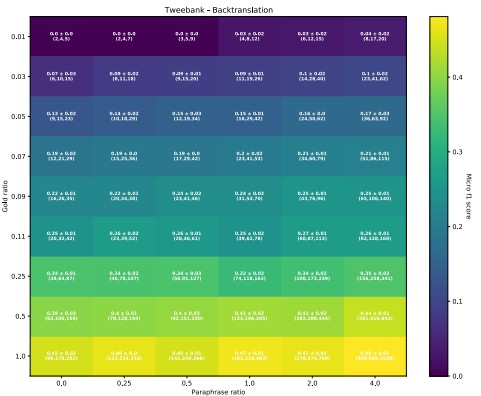

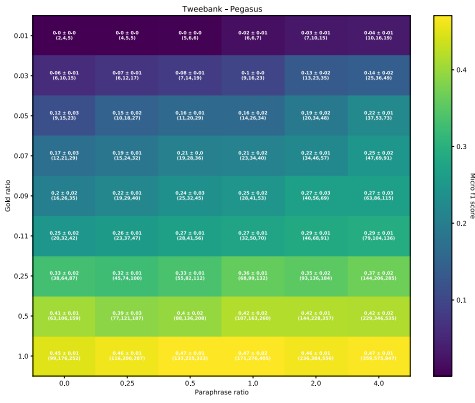

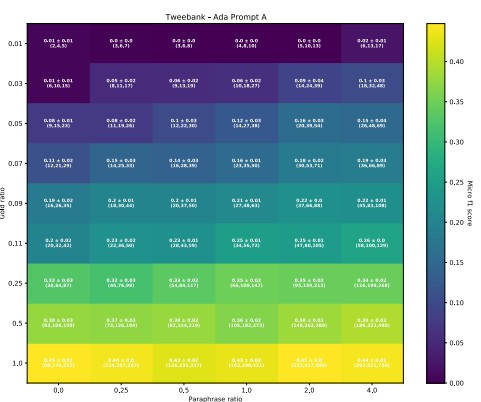

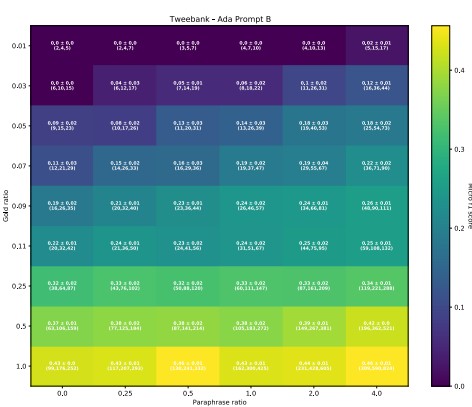

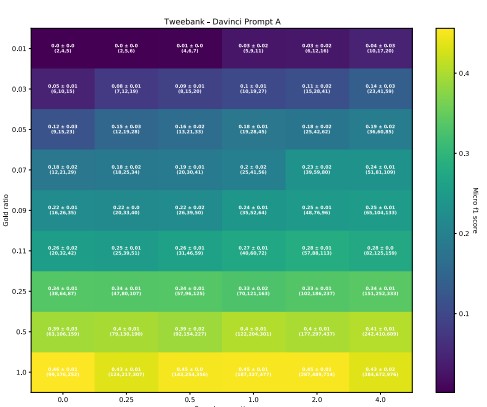

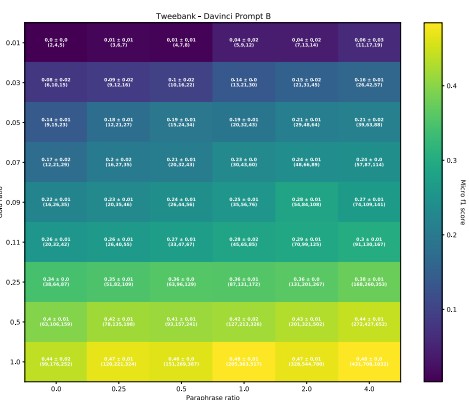

### 6.4.5 WNUT-17

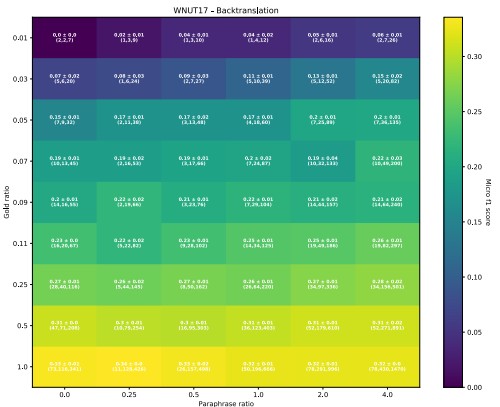
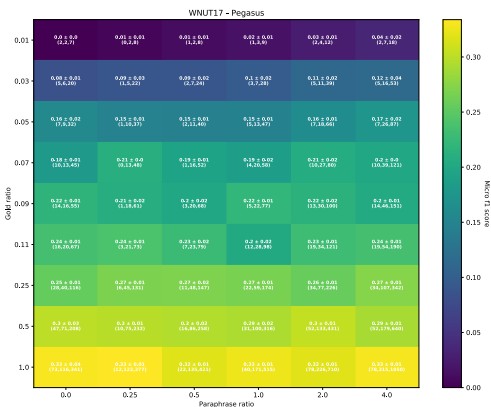

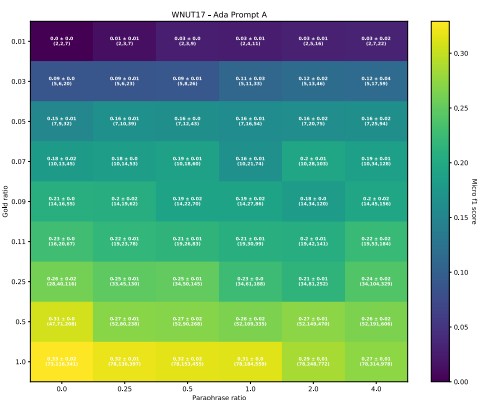
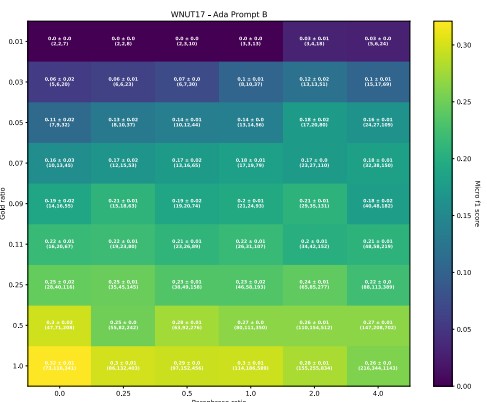

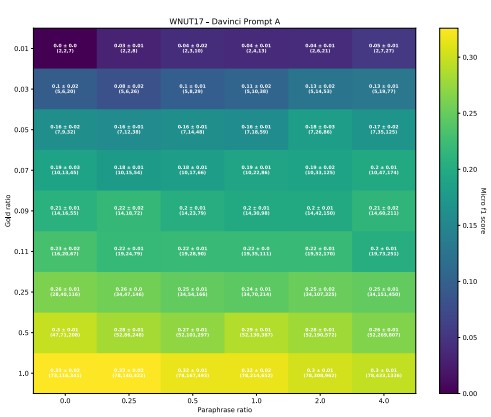
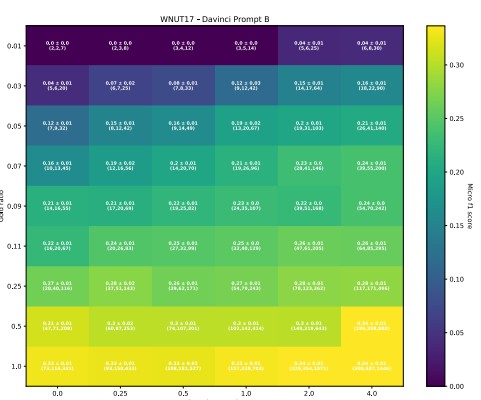

## 6.5 Human evaluation guidelines

In this document, we refer to the original sentence as "Gold" and the rephrased sentences as "Paraphrase"

We will present a set of 100 gold / paraphrase pairs from each dataset and ask annotators to annotate some metrics:

**Example**

One labeling example may look like:

**Gold**: *I am looking to invest in [Apple inc](ORG) and [TSLA](ORG)*

**Paraphrase**: *I am looking to buy [Apple](ORG) stock and [AMZN](ORG) stock.*

**Entity specific metrics:**

- How many entities (irrespective of type) in gold, are absent from paraphrase? (Fn) → 1 → TSLA*
- How many entities in gold are present in paraphrase and also annotated with correct type? (Tp) → 1 (Apple)*
- How many entities in paraphrase are absent in gold, but correct? → eg. 1 → AMZN**
- How many entities in paraphrase have wrongly annotated span?
- How many entities in paraphrase have wrongly annotated type?

For empty paraphrases, please consider them legitimate paraphrases, and annotate as appropriate. eg. all gold entities would be missing from an empty paraphrase.

*Notice we do not care for using an equivalent name/phrase for gold entity. eg. "nearby" is the same as "close by"; "Apple inc" is the same as "AAPL" etc.*

**Hallucination*

**Quality metrics:**

- Score paraphrases on a scale of 1-5. 1 being worst, 5 being the best.

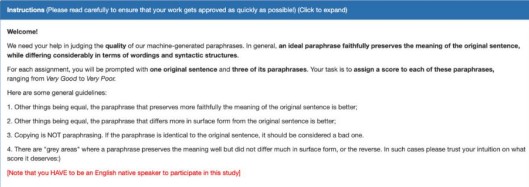

Figure 4: Interface of our MTurk studies for head-to-head comparisons with other models.

- 
- There can be ties / same score for multiple paraphrases
- Ignore annotation → Only look at the actual sentence.
  - eg. in this→ *"I am looking to buy [Appl](ORG)e stock and [AMZN](ORG) stock.";* only consider the text → *"I am looking to buy Apple stock and AMZN stock"*

## 6.6 Software Acknowledgements

This work would be much harder without the use of several software packages including, but not limited to Pytorch (Paszke et al., 2019), Huggingface transformers (Wolf et al., 2020), Scipy (Virtanen et al., 2020), Pandas (McKinney et al., 2011), Numpy (Harris et al., 2020), Scikit-learn (Pedregosa et al., 2011), and OpenAI models and Python library.

