# OpenReview forum: "Systematic review of effect of data augmentation using paraphrasing on Named entity recognition"
_NeurIPS.cc/2022/Workshop/SyntheticData4ML — Neurips 2022 SyntheticData4ML_

### Official Review · Reviewer_ttue · 2022-10-15
**Useful investigation of using paraphrasers for data augmentation for the particular task of NER**

**Rating:** 7
**Confidence:** 3

**Review:**

The paper was very well written and easy to follow. However, Figure 1 was a bit difficult to interpret. Would make more sense for the cell colours to correspond to the value, rather than paraphrasers. Same with Figure 2. Would suggest listing values for a particular dataset/paraphraser together. A wide range of datasets and paraphrasers were investigated, presenting useful information for selecting paraphrasers for future tasks.

---

### Official Review · Reviewer_MtJG · 2022-10-17
**Accept but need to work on clarity**

**Rating:** 6
**Confidence:** 4

**Review:**

### Summary
The paper presents a GPT-3 prompt-based approach for generating paraphrases while preserving entity span. By augmenting the original golden training data with the generated paraphrases, the authors have shown the prompt-based approaches outperform other state-of-art paraphrasers across different scenarios of golden annotated data ratio and paraphrase ratio for downstream NER applications.

### strength of paper
1. The improvement of NER accuracy via GPT-3 is significant compared to other paraphrasers
2. The experiments featuring different paraphrasers and downstream applications for rigorous comparison
3. The experiments are thorough and support their hypothesis

### weakness of paper
1. Using GPT-3 for paraphrasing is not a new idea. Although the authors have compared two different GPT-3 variants, it is not clear if this prompt chosen is optimal. For example, prompt tuning can potentially address this issue https://aclanthology.org/2021.emnlp-main.243/
2. The clarity of the paper can be improved. The design of the golden ratio is not intuitive, requires extra efforts to understand which one corresponding to the low data regimes. Maybe explain it better in the text or use a different visualization approach.
3. The visualization in Figure 1 and Figure 2 requires extra efforts for the reader to find evidence to support the conclusion. Consider replace the figures with tables or graphs.

### minor points
1. It would be great to see some example outputs from A and B approach for an illustration why no correlation between the paraphrase quality and downstream performance. For example, Figure 3(b) shows DaVinciA is better than DaVinciB for human evaluation except for the tweebank dataset, while DavinciB consistently perform better than DaVinciA in Figure 3(a) in terms of Entity-Recall.
2. The abstract requires some work to highlight the contribution.

---

### Official Review · Reviewer_P2WQ · 2022-10-18
**Interesting technique but lacking in**

**Rating:** 5
**Confidence:** 4

**Review:**

The presented work is interesting with some novelty, mostly in specific ways the paraphrasing is used in the context (systematic investigation of the effects of different levels of
44 paraphrasing on downstream performance, at different levels of gold annotations across 5 datasets).
It is a well-written paper with well-organized and clearly organized presentation of results.
The two major deficiencies of the paper are:

* the presented research does not seem scientifically significant. It is mostly a report on the comparative performance
* one of the major approaches used is a GPT-3 paraphraser API which is a limited access, “canned” API.

For these reasons, I do not recommend this paper for the workshop. It would need a major revision to the research approach and it’s novelty.

---

### Meta-Review · Area_Chair_2fYU · 2022-10-19

**Recommendation:** Accept

**Review:**

The authors empirically assess how paraphrasing can help NER. Even though it is mostly empirical, the paper tackles an interesting problem while benchmarking multiple methods such as GPT3, and the experimental section seems to be thoroughly constructed, which leans me to recommend an accept.